# Primate lentiviruses require Inositol hexakisphosphate (IP6) or inositol pentakisphosphate (IP5) for the production of viral particles

**Clifton L. Ricana**[1], **Terri D. Lyddon**[1], **Robert A. Dick**[2], **Marc C. Johnson**[1]*

**1** Department of Molecular Microbiology and Immunology, Bond Life Sciences Center, University of Missouri, Columbia, Missouri, United States of America, **2** Department of Molecular Biology and Genetics, Cornell University, Ithaca, New York, United States of America

* marcjohnson@missouri.edu

**Data Availability Statement:** All data files for "IP 33% PAGE Gels Images & Analysis" are available from the Harvard Dataverse Network database (https://doi.org/10.7910/DVN/05KYQZ). All "Rcode

## Abstract

Inositol hexakisphosphate (IP6) potently stimulates HIV-1 particle assembly *in vitro* and infectious particle production *in vivo*. However, knockout cells lacking inositol-pentakisphosphate 2-kinase (IPPK-KO), the enzyme that produces IP6 by phosphorylation of inositol pentakisphosphate (IP5), were still able to produce infectious HIV-1 particles at a greatly reduced rate. HIV-1 *in vitro* assembly can also be stimulated to a lesser extent with IP5, but until recently, it was not known if IP5 could also function in promoting assembly *in vivo*. Here we addressed whether there is an absolute requirement for IP6 or IP5 in the production of infectious HIV-1 particles. IPPK-KO cells expressed no detectable IP6 but elevated IP5 levels and displayed a 20-100-fold reduction in infectious particle production, correlating with lost virus release. Transient transfection of an IPPK expression vector stimulated infectious particle production and release in IPPK-KO but not wildtype cells. Several attempts to make IP6/IP5 deficient stable cells were not successful, but transient expression of the enzyme multiple inositol polyphosphate phosphatase-1 (MINPP1) into IPPK-KOs resulted in near ablation of IP6 and IP5. Under these conditions, we found that HIV-1 infectious particle production and virus release were essentially abolished (1000-fold reduction) demonstrating an IP6/IP5 requirement. However, other retroviruses including a Gammaretrovirus, a Betaretrovirus, and two non-primate Lentiviruses displayed only a modest (3-fold) reduction in infectious particle production from IPPK-KOs and were not significantly altered by expression of IPPK or MINPP1. The only other retrovirus found to show a clear IP6/IP5 dependence was the primate (macaque) Lentivirus Simian Immunodeficiency Virus, which displayed similar sensitivity as HIV-1. We were not able to determine if producer cell IP6/IP5 is required at additional steps beyond assembly because viral particles devoid of both molecules could not be generated. Finally, we found that loss of IP6/IP5 in viral target cells had no effect on permissivity to HIV-1 infection.

& Data Files for Figure Generation" are available from the Harvard Dataverse Network database (https://doi.org/10.7910/DVN/KAVRLD). All data files for "Sequence Verification of KOs & MSA for Retroviruses" are available from the Harvard Dataverse Network database (https://doi.org/10.7910/DVN/KWTOJ2). All data files for "Western Blot Images & Analysis" are available from the Harvard Dataverse Network database (https://doi.org/10.7910/DVN/YC4PS4). All data files for "EM Images & Analysis" are available from the Harvard Dataverse Network database (https://doi.org/10.7910/DVN/3NM5JB).

**Funding:** This work was supported by the National Institute of Allergy and Infectious Diseases (NIAID; https://www.niaid.nih.gov) under awards R21AI143363 to Marc C. Johnson, R01AI147890 to Robert A. Dick. Rob A. Dick performed work in lab funded by NIAID grant R01AI150454, awarded to Volker M. Vogt. The funders had no role in study design, data collection and analysis, decision to publish, or preparation of the manuscript.

**Competing interests:** The authors have declared that no competing interests exist.

## Author summary

Inositol hexakisphosphate (IP6) is a co-factor required for efficient production of infectious HIV-1 particles. The HIV-1 structural protein Gag forms a hexagonal lattice structure. The negatively charged IP6 sits in the middle of the hexamer and stabilizes a ring of positively charged lysines. Previously described results show that depletion of IP6 reduces, but does not eliminate, infectious virus production. This depletion was achieved through knock-out of inositol-pentakisphosphate 2-kinase (IPPK-KO), the enzyme responsible for adding the sixth and final phosphate to the molecule. Whether IP6 is absolutely required, another inositol phosphate can substitute, or IP6 is simply acting as an enhancer for virus production was unknown. Here, we show that loss of both IP6 and inositol pentakisphosphate (IP5) abolishes infectious HIV-1 production from cells, demonstrating that at least one of these molecules is required during the assembly process. We do this through a cell-based system using transiently expressed cellular proteins to restore or deplete IP6 and IP5 in the IPPK-KO cell line. We further show that the IP6 and IP5 requirement is a feature of primate lentiviruses, but not all retroviruses, and that IP6 or IP5 is required in the producer but not the target cell for HIV-1 infection.

## Introduction

The HIV-1 structural protein Gag is produced in the cytoplasm and traffics to the plasma membrane where it assembles into a viral particle that buds from the host membrane [1]. Gag is a polyprotein consisting of the Matrix (MA), Capsid (CA), Spacer 1 (SP1), Nucleocapsid (NC), Spacer 2 (SP2), and p6 domains [1,2]. During assembly, the Gag protein assembles into an 'immature' hexagonal lattice, driven primarily by interactions involving the CA and SP1 domains [1,3]. The C-terminal CA and SP1 domains contain an alpha-helix that forms a six-helix bundle with the other Gag proteins in the hexamer [4,5]. This bundle is important in formation and stabilization of the immature lattice [4,5]. During or shortly after budding from the cell, the viral protease cleaves the Gag polyprotein into its constitutive components, which separates CA from SP1 and eliminates the six-helix bundle [1,6]. The liberated CA protein then assembles into a structurally distinct 'mature' lattice, which forms the viral core [1,7].

Early attempts to assemble full length HIV-1 Gag protein *in vitro* revealed that proper assembly required the presence of cell lysate [8]. This pointed to an assembly co-factor that catalyzed viral assembly in cells. Further research revealed that inositol phosphates were sufficient to stimulate proper assembly, but the mechanistic basis for this effect was poorly understood [3]. Recently, a Cryo-EM reconstruction of *in vivo*-produced immature HIV-1 particles revealed a small density above the CA-SP1 six-helix bundle that was coordinated by two rings of lysine residues, suggesting the presence of a negatively charged molecule inside the particle that helped stabilize the bundle [4]. This evidence for such a molecule, in conjunction with previous data that inositol phosphates stimulate assembly, prompted further evaluation of the role of inositol phosphates as HIV-1 assembly co-factors [3,8,9]. In particular, Inositol hexakisphosphate ($I(1,2,3,4,5,6)P_6$ or IP6), which is a hexagonal six-carbon ring with a negatively charged phosphate at each position, seemed like a likely match for the density identified in particles [10].

In assembly experiments *in vitro*, the presence of IP6 was found to potently promote immature assembly and even to modulate whether particular Gag proteins assembled into immature or mature lattices [10]. Mutation of the lysine residues in Gag believed to coordinate the negatively charged molecule made the Gag proteins 100-fold less responsive to IP6 in *in vitro*

assembly reactions [10]. When a crystal structure of the HIV-1 CA$_{CTD}$SP1 protein in the presence of IP6 was solved, a density was observed associated with the six-helix bundle that precisely matched the density described in the *in vivo* Cryo-EM reconstruction [4,10]. These biochemical and structural data strongly support the conclusion that the density observed in HIV-1 particles is indeed IP6, but the data could not reveal whether IP6 is an absolute requirement for HIV-1 assembly *in vivo*.

IP6 is found in mammalian cells at concentrations of 10-100uM [11], and is synthesized by a series of host enzymes through a complex and not fully resolved process (Fig 1A) [12–28]. The immediate precursor to IP6 is inositol pentakisphosphate (I(1,3,4,5,6)P$_5$ or IP5), and the only enzyme known to catalyze the addition of the final 2-phosphate is inositol-pentakisphosphate 2-kinase (IPPK) [12–16]. IP5 was also shown to stimulate immature HIV-1 assembly *in vitro*, though not as robustly as IP6 [10,29–31]. In cells, several pathways have been described that lead to IP5 production, but all of those described in mammalian cells require the enzyme inositol-polyphosphate multikinase (IPMK) [12,16–20]. However, cells derived from a homozygous mouse embryo deficient in IPMK still produced residual levels of IP5 and IP6 through an unknown mechanism [15,17]. Recently, a genetic screen performed to identify genes involved in necroptosis revealed that inositol phosphates IP5 or IP6 are required for this process [21]. Importantly, the screen identified the genes *IPMK* and *inositol-tetrakisphosphate 1-kinase (ITPK1)* as being required for necroptosis, and cells lacking either of these genes were noticeably deficient in IP5 and IP6 [21,22]. Thus, IPMK and ITPK1 likely cooperate in the production of IP5 in cells. Moreover, knockouts of the IPPK and IPMK genes have both been reported to reduce, but not abolish, the production of infectious HIV-1 particles *in vivo* [10,29–31].

Here, we sought to determine whether the presence of inositol phosphates IP6/IP5 are an absolute requirement for HIV-1 particle. To accomplish this, we developed a system to transiently deplete cells of both IP6 and IP5 and test the assembly competence of HIV-1 and various other retroviruses under these conditions. During the preparation of this manuscript, Mallery *et al.* [31] published findings that IP5 can substitute for IP6 without loss of infectivity of the particles produced. In addition to their findings, we found that HIV-1 is absolutely dependent on the presence of IP6 or IP5 for viral production, but non-primate lentiviruses and viruses from retrovirus genera other than lentivirus are not. We further found that neither IP6 nor IP5 is required in viral target cells for successful HIV-1 infection.

## Results

### IPMK contributes to IP6 synthesis and infectious virus production

We previously showed that IP6 stimulates immature *in vitro* HIV-1 particle assembly [10]. We further showed that knock out of IPPK, the only enzyme known to catalyze addition of the final phosphate in the generation of IP6 (Fig 1A and 1B) [12–16], resulted in a drastic reduction in infectious HIV-1 particle production from cells [10]. However, infectious particles were still produced from HEK293FT IPPK knockout (IPPK-KO) cells, albeit at a greatly reduced rate [10]. There are three possible explanations for this partial phenotype. First, the cells could be continuing to produce low levels of IP6 through an unknown mechanism. Second, IP6 may enhance infectious HIV-1 particle production, but not strictly be required for it. Finally, IP5, the precursor to IP6 that can partially stimulate HIV-1 particle production *in vitro*, could substitute for IP6 in infectious particle production. To test the latter explanation, we attempted to generate a cell line that is deficient in both IP5 and IP6 production, in order to test if such a cell line would be completely deficient in infectious particle production. The enzyme inositol-polyphosphate multikinase (IPMK) has been reported to catalyze the

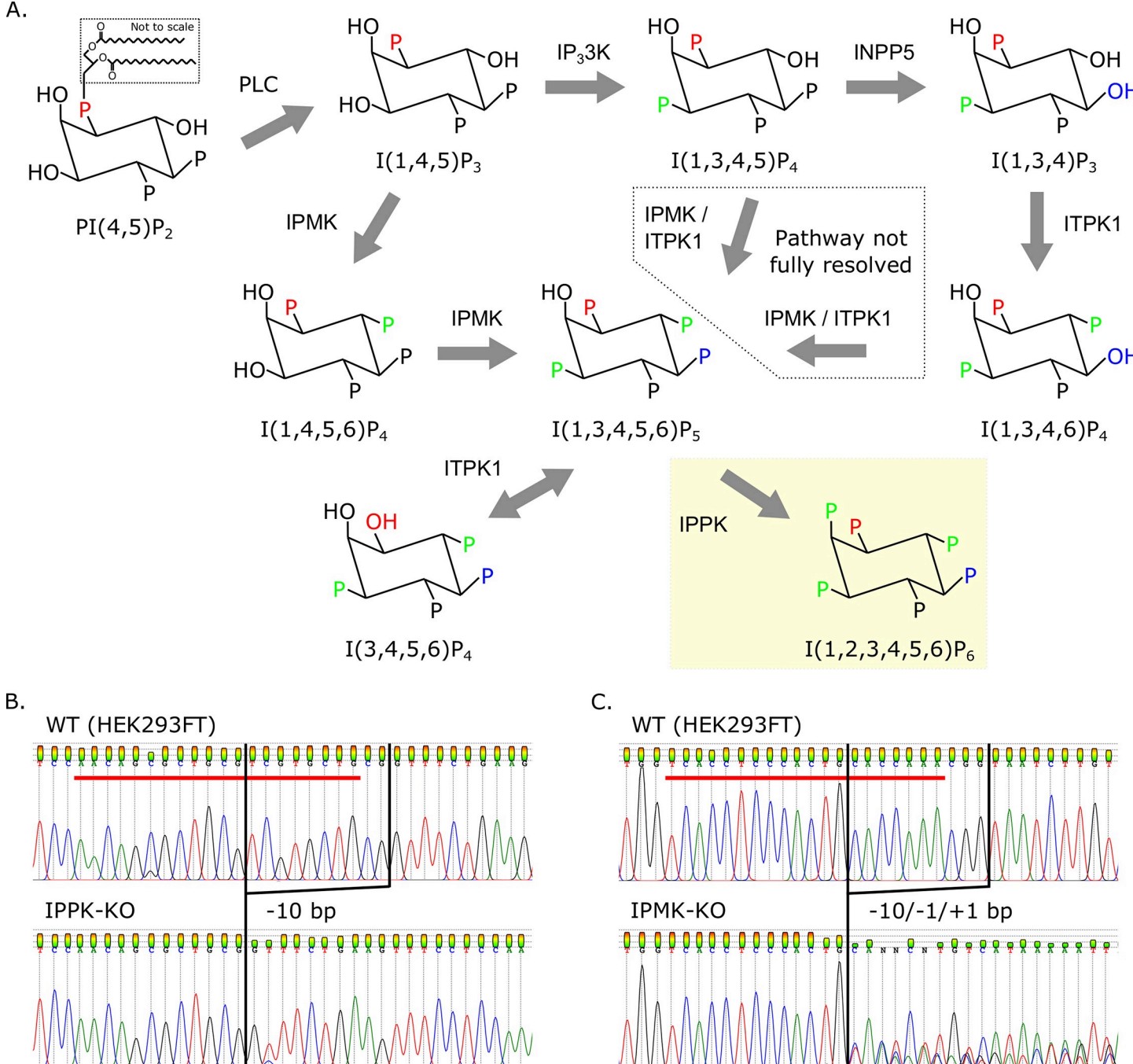

**Fig 1. Knock-out of cellular genes leading to the production of IP6.** (A) Inositol phosphate pathway in *H. sapiens*. Inositol-pentakisphosphate 2-kinase (IPPK) adds the sixth phosphate to position 2 of IP5 (yellow box). IP5 synthesis from I(1,3,4,5)P$_4$ and I(1,3,4,6)P$_4$ has not been fully resolved. Other abbreviations: phospholipase C (PLC), IP$_3$3K (inositol-triphosphate 3-kinase), IPMK (inositol-polyphosphate multikinase), INPP5 (inositol-polyphosphate 5-phosphatase), and ITPK1 (inositol-tetrakisphosphate 1-kinase). (B-C) Chromatograms showing insertion-deletions of inositol-phosphate pathway KOs in HEK293FTs. Red bars delineate the 20-base pair guide RNA sequence used for CRISPR/Cas9 targeting. (B) KO of IPPK has a 10-base pair (bp) deletion. (C) KO of IPMK has three copies with 1- and 10-bp deletions and a 1-bp insertion.

penultimate step in IP6 production; thus, knockout of this enzyme would theoretically abolish IP5 and IP6 synthesis [12,16–22]. We used CRISPR/Cas9 with a guide RNA against *IPMK* [32,33] to generate an isolated clonal IPMK knockout (IPMK-KO) cell line (Fig 1C). The

validated clonal IPMK-KO cell line was then compared to HEK293FT cells and the previously described IPPK-KO cell line (previously described in [10]; sequence validations for selected clones shown in Fig 1B and 1C).

First, we wanted to validate the loss of IP6 and IP5 in our IPPK-KO and IPMK-KO cells respectively. Using $TiO_2$ extraction and 33% PAGE separation [34,35], we found that the IPPK-KO cells had no detectable IP6, but a slightly elevated level of IP5 compared to HEK293FT cells (Fig 2A–2C). From a functional standpoint, this confirms our KO. In contrast, the IPMK-KO cells had residual levels of both IP6 and IP5 (Fig 2A–2C). This finding is consistent with a previous report that showed that cells from an IPMK knockout embryo were also shown to produce low levels of IP5 and IP6 through an unknown mechanism [21,22].

Next, we measured infectious HIV-1 particle production from HEK293FT cells and its two derivative knockout lines. An HIV-1$^{\Delta Env}$ provirus containing a GFP reporter (HIV-CMV-GFP) was co-transfected with a VSV-G expression construct into the three cell lines in parallel,

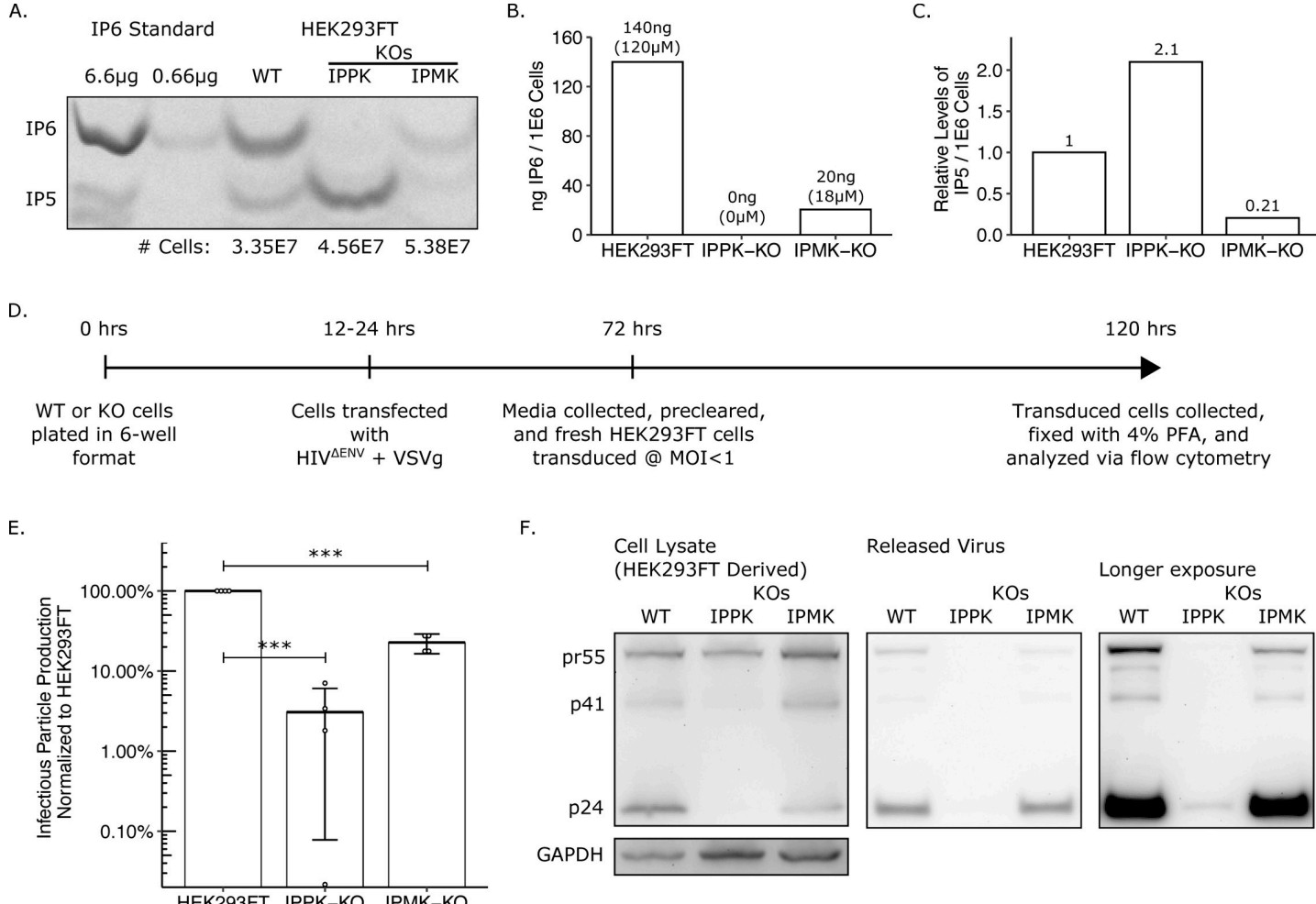

**Fig 2. IP pathway KOs have reduced IP6 and IP5 levels and have a loss of infectious particle release.** (A) 33% PAGE gel separating inositol phosphates. Two dilutions of purified 1 M IP6 were used as a standard and had IP5 breakdown products. The number of cells in each sample is indicated. (B) IP6 quantification of panel A in ng per million cells and μM. (C) Relative IP5 quantification normalized to the HEK293FT control. (D) Experimental timeline. (E) Percent infectious particle release normalized to HEK293FT cells. Student's t-test was used for pair-wise comparison (n = 4, *** p < 0.001, error bars = mean + SD). (F) Representative western blot of experiments from panel D. Full-length HIV-Gag (pr55) and GAPDH loading control are presented on the left panels. Virus released into media is presented in the middle panel. A longer exposure of the virus release blot was also taken and presented on the right panel.

and the media was titered on fresh target cells (Fig 2D). As we reported previously, infectious particle production from the IPPK-KO cell line was reduced ~20–100 fold compared to HEK293FT cells (Fig 2E) [10]. The IPMK-KO cells displayed a more modest ~5-fold reduction in infectious particle production that corresponded with the residual IP6 levels found in the cells (Fig 2E). We then tested whether the block in infectious virus particle production was due to a block in virus release or to reduced infectivity of released virus. To accomplish this, we measured the pr55 Gag/p24 CA protein level in producer cells and the supernatant. Western blots with an antibody against p24 CA revealed that HEK293FT, IPPK-KO, and IPMK-KO cells all produced full length pr55 Gag at relatively equal levels (Fig 2F, left). However, p24 CA released into the supernatant was barely detectable from the IPPK-KO cells, while media from IPMK-KO cells contained normal p24 levels (Fig 2F, middle and right). If pr55 Gag is not released as virus particles, one might expect to see a buildup of Gag in transfected cells. In the representative blot shown, pr55 Gag in the IPPK-KO cells was slightly reduced relative to GAPDH. However, because some viral transduction occurs within the transfected cells that amplifies the Gag production, this could explain why cells that are able to support infectious particle production appear to express slightly higher Gag levels. Together, the western blot data show that loss in infectious particle production primarily correlates with a loss in viral release.

## Exogenous addition of MINPP1 depletes IP6 and IP5 and is toxic to cells

Knock-out of IPPK ablates IP6 in cells while slightly increasing IP5 levels, while knock-out of IPMK leaves residual levels of IP6 and IP5. The finding that low-level infectious particle production still occurs in IPPK-KO cells is consistent with the hypothesis that IP5 can substitute for IP6 in supporting HIV-1 assembly. However, because neither IPPK-KO nor IPMK-KO cells were completely devoid of IP5 and IP6, we could not rule out other explanations. Therefore, we next attempted to make pairwise knockouts of IPPK, IPMK, and ITPK1 (which also contributes to IP5 synthesis [12,16,21–27]) to further reduce inositol phosphate levels. Numerous attempts failed to yield such double knockouts, likely because loss of the combination of enzymes was lethal. Therefore, as an alternative approach, we attempted to modulate levels of inositol phosphates by overexpressing multiple inositol polyphosphate phosphatase-1 (MINPP1), an enzyme that removes the 3-phosphate from IP6 and IP5 (Fig 3A) [12,36–38]. While removal of 3-phosphate from IP6 results in the production of an alternative species of IP5 (I(1,2,4,5,6)P$_5$), this IP5 species has an equatorial hydroxyl group that is likely not favorable for interaction with the lysines in the IP6 binding pocket (Fig 3A).

We first tested whether inositol phosphates could be modulated in a transient assay. To do this, we generated expression constructs containing cDNAs for IPPK or MINPP1 on a plasmid that also expressed a selectable hygromycin gene. These vectors were individually transfected into HEK293FT cells or IPPK-KO cells, the cells were briefly treated with hygromycin to eliminate untransfected cells, and inositol phosphate levels were directly measured from the surviving cells (Fig 3B). In HEK293FT cells, exogenous expression of IPPK increased IP6 levels while decreasing IP5 levels (Fig 3C–3E, column 2). In contrast, exogenous expression of MINPP1 reduced, but did not ablate, IP6 and IP5 levels (Fig 3C–3E, column 3). IPPK-KO cells had no detectable IP6, but expression of IPPK restored IP6 levels (Fig 3C–3E, columns 4–5). Expression of MINPP1 in IPPK-KO cells resulted in a near complete loss of IP5 in addition to the loss of IP6 (Fig 3C–3E, column 6). This result demonstrates that exogenous expression of MINPP1 is a viable method of modulating intracellular IP5 levels.

To reduce the inherent variability in transient transfection experiments, we attempted to stably express MINPP1 in IPPK-KO cells. These attempts to stably express MINPP1 resulted

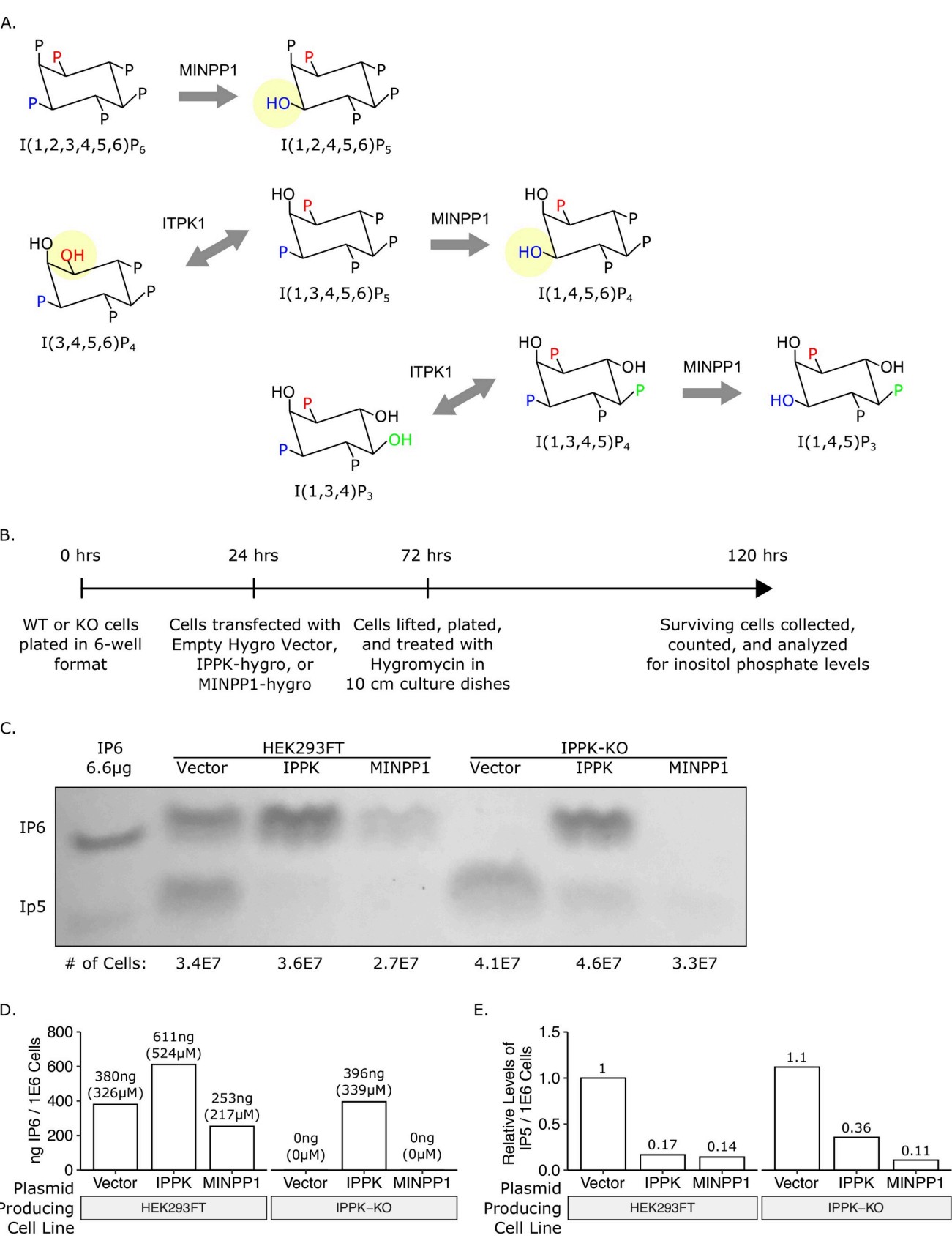

**Fig 3. Addition of multiple inositol polyphosphate phosphatase 1 (MINPP1) removes endogenous IP6 and relevant IP5 species from cells.** (A) Inositol phosphate pathway showing MINPP1 removal of the 3-position phosphate from IP6, IP5, and IP4. Removal of 3-phosphate from IP6 and I $(1,3,4,5,6)P_5$ results in an equatorial hydroxyl group. (B) Experimental timeline. (C) 33% PAGE gel separating inositol phosphates. Dilution of purified 1 M IP6 was used as a standard and had an IP5 breakdown product. The number of cells in each sample is indicated. (D) IP6 quantification of panel C in ng per million cells and μM. (E) Relative IP5 quantification of panel C normalized to the HEK293FT control.

in poor recovery under hygromycin in IPPK-KO cells but was tolerated in HEK293FT cells. Previous reports indicated that MINPP1 expression can induce apoptosis [36–38]. To verify that the low recovery of MINPP1 expressing cells was in fact due to toxicity, we created a retroviral reporter vector in which MINPP1 cDNA was followed by an IRES-EGFP (Fig 4A). Cells transduced with this reporter should express both MINPP1 and EGFP, and cell survival can be determined by measuring the number of GFP positive cells over time (Fig 4B). HEK293FT cells expressing MINPP1-IRES-GFP or a control (IRES-GFP alone, Fig 4A) were maintained in the population over the course of three weeks indicating tolerance of MINPP1 (Fig 4C and

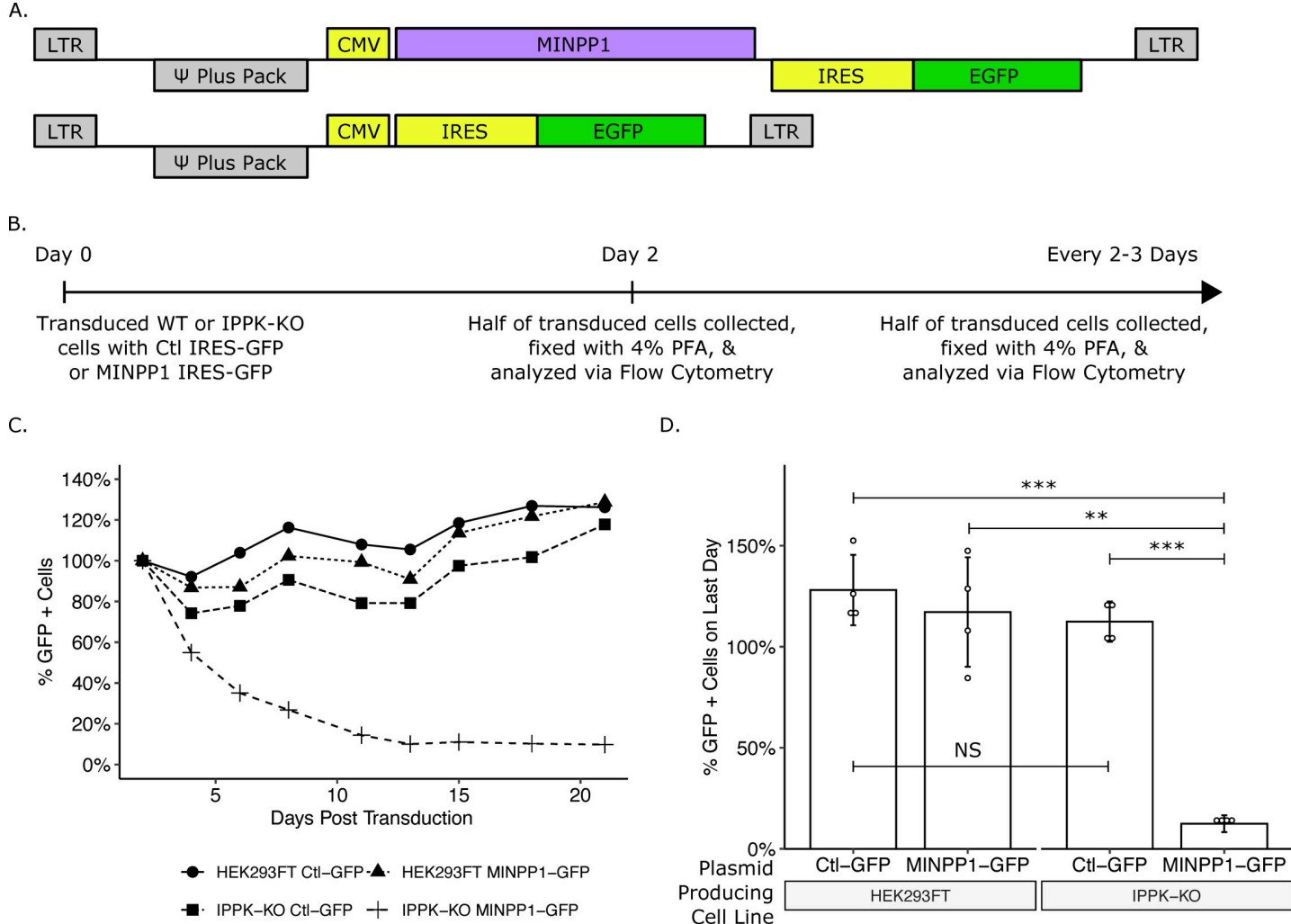

**Fig 4. Exogenous expression of MINPP1 is toxic in IPPK-KO cells.** (A) Plasmid map of expression vectors. (B) Experimental timeline. (C) Line plot of a representative experiment. The percentage of cells expressing EGFP over time are normalized to the starting population of EGFP positive cells for each cell line. (D) Bar chart of percent EGFP positive cells on the last day of collection from panel C (day 21). Student's t-test was used for pair-wise comparison (n = 4, ** p < 0.01, *** p < 0.001, error bars = mean ± SD).

4D). However, the majority of IPPK-KO cells expressing MINPP1-GFP were lost over the course of the experiment, consistent with toxicity (Fig 4C and 4D). This supports previous reports of a balance of IP6 and IP5 promoting cell viability [37,38]. Importantly, cell death from MINPP1 expression in IPPK-KO cells was not immediate, which allowed a window for testing the effects of MINPP1 expression on virus production.

## Exogenous addition of MINPP1 essentially abolishes HIV-1 infectious virus production

Because it was not possible to make a stable IPPK-KO cell line expressing MINPP1, we chose to test the effects of MINPP1 expression on virus production using a transient expression assay. Briefly, HIV-CMV-GFP and VSV-G DNAs were co-transfected into HEK293FT or IPPK-KO cells with either an expression vector containing IPPK cDNA, MINPP1 cDNA, or no insert, and the cells were allowed to produce virus for two days (Fig 5A). Infectious virus particles were then titered on HEK293FT cells (Fig 5A). As before, infectious particle production was reduced 20-100-fold from IPPK-KO cells (Fig 5B). Addition of IPPK to HEK293FT cells had no appreciable effect on this number, but addition to IPPK-KO cells enhanced infectious particle production by approximately 10-fold (Fig 5B). Likewise, addition of MINPP1 to HEK293FT cells also had no appreciable effect on infectious particle production, but addition to IPPK-KO cells further reduced infectivity by approximately 10-fold, which approached background levels (Fig 5B). Western blotting was again used to determine at which step in infectious virus particle production was blocked (Fig 5C). The expression of pr55 Gag in cells was similar among all conditions, indicating that modulation of inositol phosphate levels was not grossly affecting translation levels (Fig 5C, top and middle). Exogenous expression of IPPK and MINPP1 did not appear to affect viral release or protein maturation from HEK293FT cells (Fig 5C, bottom left). Introduction of IPPK and or MINPP1 into the IPPK-KO did not noticeably alter pr55 Gag expression levels relative to the GAPDH control (Fig 5D). However, viral release from IPPK-KO cells varied considerably across the conditions, and the amount of virus released closely tracked with infectious particle production (Fig 5C, bottom right; Fig 5E, quantification). HIV-1 CA release in IPPK-KO cells, which still express IP5, was greatly stimulated by expression of IPPK cDNA, but CA released was essentially abolished by expression of MINPP1, which removes the IP5. This is in agreement with an earlier report from Mallery et al. that showed that viral particle production was reduced in cells lacking IPPK and that the remaining infectious particles produced contained IP5 in place of IP6 [31]. This current data goes further to suggest that the presence of IP6 or IP5 is an absolute requirement for the release of viral particles. However, because there were essentially no virus particles released in the absence of IP5 and IP6, it was not possible to determine if any such particles might have been infectious. Thus, it remains possible that IP5 and/or IP6 also are required at other stages of the viral life cycle.

## Depletion of IP6 and IP5 in target cells does not affect susceptibility to HIV-1 infection

IP6 has been shown to stabilize the lattice of the immature and mature hexamer [10,29–31]. This stabilization has been proposed to be important for DNA synthesis following the release of the viral core into the cytoplasm of target cells [29]. When the viral core is depleted of IP6 *in vitro*, it breaks down more readily. Thus it has been inferred that after fusion with the target cell, IP6/IP5-lacking virus cannot effectively reverse transcribe the viral RNA to DNA [29,39]. To test if IP6 in the target cell is required for susceptibility to infection, we infected HEK293FT or IPPK-KO cells with virus produced from either HEK293FT or IPPK-KO cells and

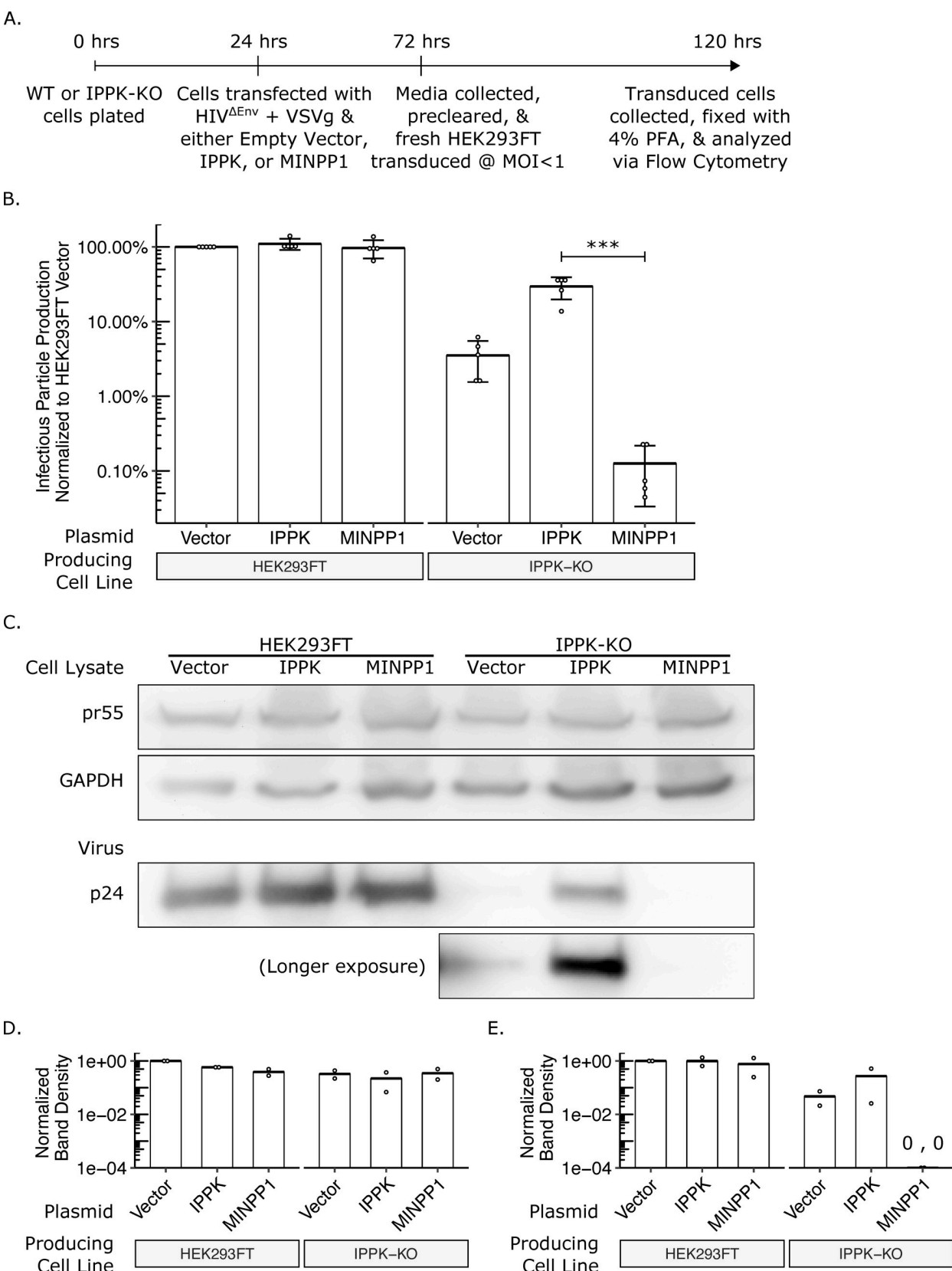

**Fig 5. IPPK-KO cells expressing MINPP1 have substantial loss in infectious particle production due to a block in viral release.** (A) Experimental timeline. (B) Bar chart of percent infectious particle release normalized to virus from HEK293FT cells expressing the empty vector. Student's t-test was used for pair-wise comparison (n = 5, *** p < 0.001, error bars = mean ± SD). (C) Representative western blot of two experiments from panel B. The rows are full-length HIV-Gag (top), GAPDH loading control (middle), and virus released into media (bottom). A longer exposure was also taken for the blot of released virus. (D) Relative quantification of uncleaved pr55 Gag normalized to GAPDH levels in cell lysate. (E) Relative quantification of p24 Gag in virus released into media.

compared infection levels. If IP6 in the target cell is required for viral infection, one would expect to see a lower viral titer in IPPK-KO cells, regardless of the source of virus. By contrast, if IP6 is required for infection but can be derived from the producer or target cells, one would expect the virus to have a lower relative titer on IPPK-KO cells only when the virus is produced from IPPK-KO cells. In fact, we observed no significant difference in titer on the two types of cells, regardless of the source of the virus (Fig 6A). While this experiment demonstrates that IP6 is not required in the target cell for infection, it does not address whether IP5 is perhaps able to substitute for IP6 at this stage of the infection.

To test the role of IP5 depletion in susceptibility of target cells, we next transduced the MINPP1-IRES-GFP vector or an empty IRES-GFP control (Fig 4A) into HEK293FT or IPPK-KO cells, and then tested their susceptibility to infection (Fig 6B). Two days after transduction with MINPP1 IRES-GFP or IRES-GFP, cells were transduced with VSV-G pseudotyped HIV-1$^{\Delta Env}$ virus containing a CD4 reporter (Fig 6C). Two days after virus transduction, surface CD4 was stained with an APC-conjugated antibody to score for successful virus transduction (Fig 6D). If cells expressing MINPP1 are less susceptible to infection, then the fraction of GFP-positive cells that are also APC positive (virus transduced) should be less than the fraction GFP-negative cells that are APC positive. If they are more susceptible, then the fraction of GFP-positive cells that are also APC positive should be more than the fraction GFP-negative cells that are APC positive. The expected fraction of GFP/APC double positive cells can then be calculated based on the total number of GFP and APC positive cell and compared to the actual number of GFP/APC double positive cells observed (Fig 6D, equation). Expression of MINPP1 was found not to alter the susceptibility of HEK293FT or IPPK-KO cells (Fig 6E). Together, these data suggest that neither IP6 nor IP5 from target cells is required for viral infection. As before, since we were not able to obtain virus that was devoid of IP5 and IP6, it was not possible to determine if IP5 and/or IP6 from the producer cell is required for core stability during infection.

## Beta- and Gamma-retroviruses do not require IP6 or IP5 for infectious virus production

Different retroviral species vary in Gag lattice structure and viral protein trafficking. The IP6 and IP5 requirement for assembly of other retroviral species can inform the different assembly strategies utilized by retroviruses. With HIV-1 as the model virus, we first tested outgroup retroviral species with our exogenous gene co-transfection system. With expression of IPPK and MINPP1 in HEK293FT cells, the Gammaretrovirus Murine Leukemia Virus (MLV) did not vary in viral output (Fig 7A). Infectious MLV particle production from IPPK-KO was reduced a few-fold compared to HEK293FT cells; however, this reduction could not be modulated further by addition of IPPK or MINPP1. Expression of IPPK actually caused a small but statistically insignificant reduction in infectious particle production compared to empty vector (Fig 7A). With expression of MINPP1 in IPPK-KO cells, there was no difference in virus output compared to empty vector (Fig 7A). These data suggest that the 3-fold reduction in virus particle release with MLV does not reflect a direct IP6 or IP5 requirement.

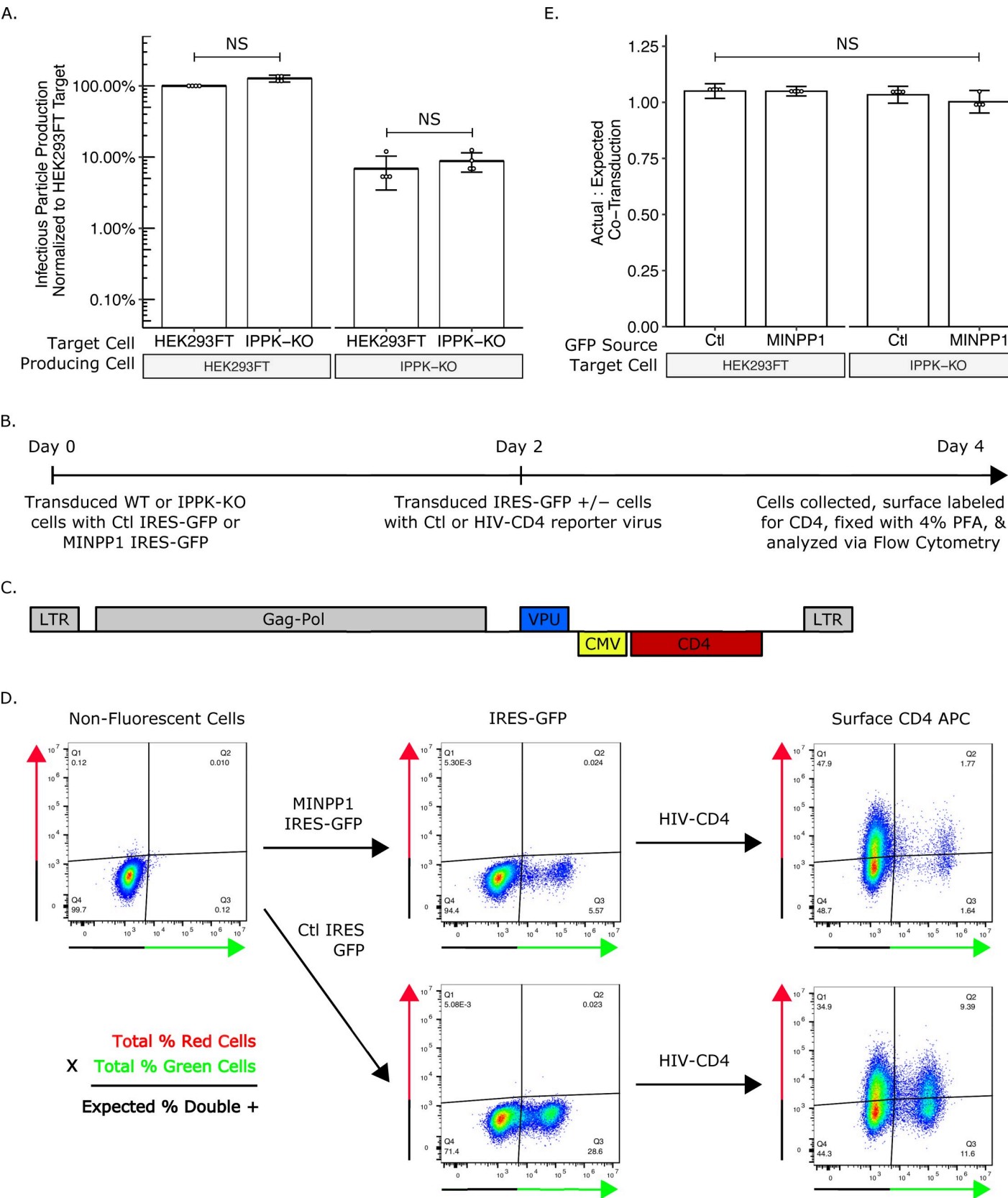

**Fig 6. IP6 and IP5 levels in target cells do not affect susceptibility to HIV-1 infection.** (A) Bar chart of percent infectious particle release normalized to HEK293FT cells. Student's t-test was used for pair-wise comparison (n = 4, * p < 0.05). (B) Experimental timeline of the assay. (C) Plasmid map of HIV-1$^{\Delta Env}$-CD4. (D) Example flow plots show output from the assay. (E) Bar chart of the ratio of the actual percent double positive cells to the expected double positive cells. The expected percent of double positive cells was calculated from the total percent of red cells and green cells. Student's t-test was used for pair-wise comparison (n = 4, * p < 0.05, error bars = mean ± SD).

We next tested the Betaretrovirus Mason-Pfizer Monkey Virus (MPMV). As with HIV-1 and MLV, expression of IPPK and MINPP1 in HEK293FT cells did not affect infectious particle release (Fig 7B). Infectious particle production was again slightly decreased from IPPK-KO cells, but neither IPPK nor MINPP1 expression altered this output (Fig 7B). As with MLV, these data suggest that MPMV does not have a strict IP6 or IP5 requirement for infectious particle production. Additionally, amino acid sequence alignment between HIV-1, MLV, and MPMV CA proteins shows no homology to the K290 and K359 residues that interact with IP6 and IP5 in HIV-1 (Fig 7C) [10].

The MLV and MPMV data suggest that while the slightly slower rate of growth of IPPK-KO cells had overall negative effects on virus output, production of Gag and other viral proteins as well as release of virus is able to proceed despite cytotoxicity, improper cellular signaling, or another cell effect. While there are cytopathic effects of IP5/IP6 depletion from cells (Fig 4C and 4D), the inability of IP5/IP6 modulation to affect virus release of MLV and MPMV suggest

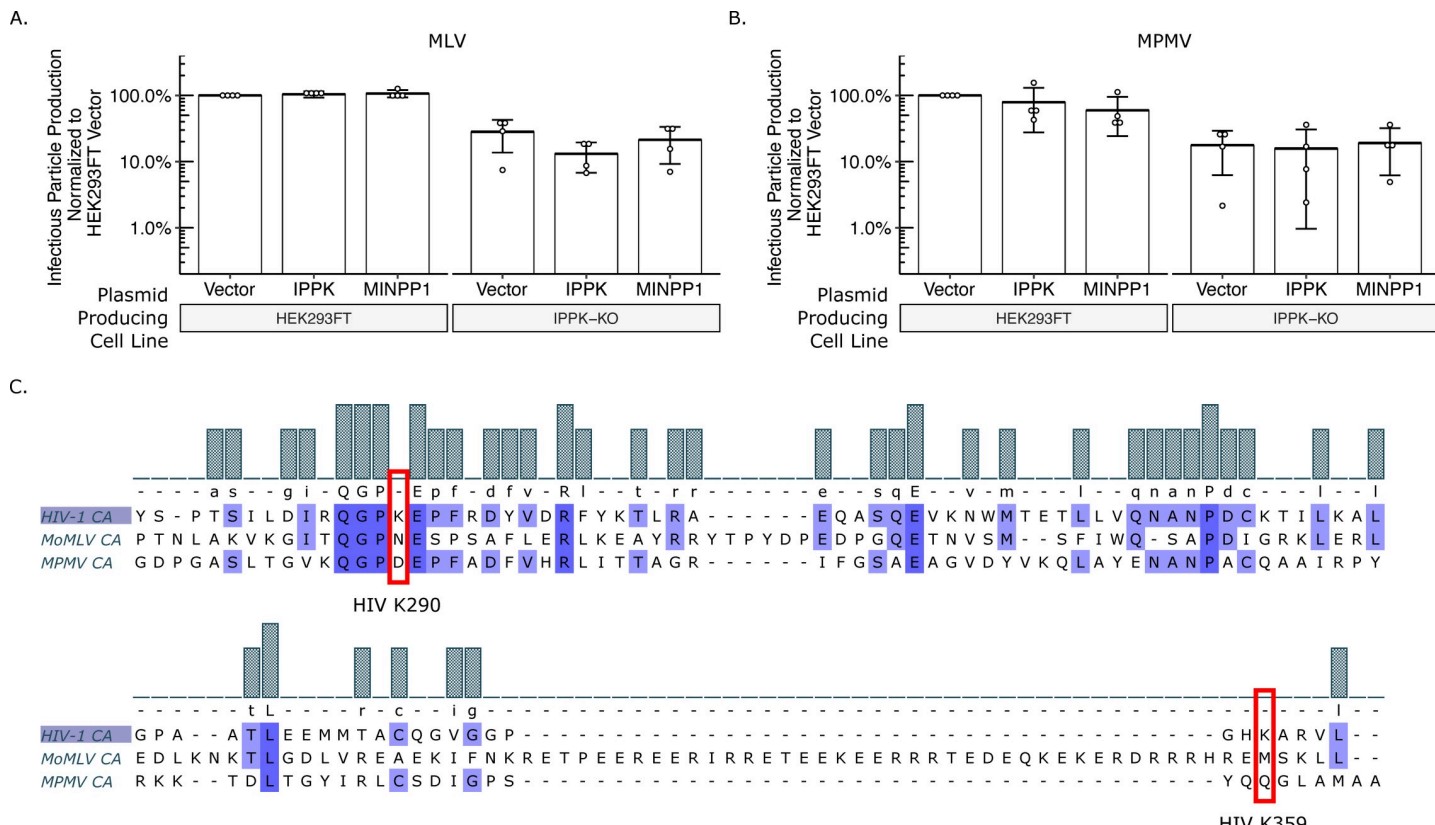

**Fig 7. Gammaretroviruses and Betaretroviruses do not require IP6 or IP5 as assembly co-factors.** Bar charts of percent infectious particle release of other retroviral genera normalized to virus from HEK293FT cells expressing the empty vector. (A) The Gammaretrovirus Murine leukemia virus (MLV, n = 4). (B) The Betaretrovirus Mason-Pfizer monkey virus (MPMV, n = 5). (C) Multiple sequence alignment of CA proteins of HIV-1, MLV, and MPMV. Note the lack of K290 and K359 homology in MLV and MPMV.

that the assembly deficiency seen in HIV was due primarily to the absence of the IP5/IP6 from the virion.

## The IP6 and IP5 requirement is conserved across primate lentiviruses

Both outgroups tested are from different retroviral genera than HIV-1. To determine whether IP6 and IP5 are required for other members of the Lentivirus genus, we next tested the primate (macaque) Simian Immunodeficiency Virus (SIV-mac), the feline Feline Immunodeficiency Virus (FIV), and the equine Equine Infectious Anemia Virus (EIAV). In our exogenous gene expression system, SIV had similar outputs to HIV-1 (Fig 8A). Infectious particle production was reduced over 20-fold from IPPK-KO cells relative to HEK293FT cells. Importantly, infectious particle production was partially restored with addition of IPPK and precipitously reduced by the addition of MINPP1. As with MLV and MPMV, transfection of IPPK-KO with EIAV and FIV produced about 3-fold fewer infectious virus particles than HEK293FT cells (Fig 8B and 8C). However, neither EIAV nor FIV were significantly affected by introduction of IPPK or MINPP1 (Fig 8B and 8C). Comparison of the protein sequence alignments shows homology between all four lentiviruses at K290 and K359; however, prolines upstream and downstream of K290 are not conserved for FIV and EIAV (Fig 8D). Together, these data point toward an IP6 and IP5 requirement for primate lentiviruses but not lentiviruses of other species.

We next wanted to determine if the sensitivity observed in infectious particle production is reflected in an assembly assay *in vitro* with purified proteins. We have reported previously that IP6 stimulates assembly of EIAV particles, despite the lack of dependence in infectivity assays [40]. In the case of EIAV, IP6 acts as an enhancer to virus particle release but is not a requirement. We therefore chose to test the stimulation of HIV-1, SIV, FIV, and EIAV in *in vitro* assembly reactions at pH8 and different IP6 concentrations (Fig 9). As expected, addition of as little as 5 μM IP6 stimulated robust assembly of immature, spherical virus like particles (VLPs) for HIV-1 and SIV-mac (Fig 9A, 9B and 9E). However, FIV and EIAV required higher concentrations of IP6 and showed more moderate effects (Fig 9C–9E), consistent with our previous findings [40]. Interestingly, the construct used for EIAV assembly in the absence of IP6 predominantly forms narrow tubes, which we previously showed to be immature-like lattices [40]. However, in the presence of IP6, they formed predominantly spherical VLPs (Fig 9D and 9F). Together, these data suggest that IP6 is a requirement for primate lentiviruses and likely acts as an enhancer to promote non-primate lentivirus assembly.

## Discussion

### The requirement of IP6 and IP5 for HIV-1 assembly *in vivo*

IP6 has been shown to be an HIV-1 assembly co-factor. The importance of IP6 has been described in HIV-1 assembly, where it promotes both immature Gag and mature CA assembly, as well as during viral entry, where it stabilizes the capsid en route to the nucleus [10,29,30]. Recently, Mallery *et al.* demonstrated that IP5 is incorporated into viral particles from cells deficient in IP6 production [31]. While release of virus from their IPPK-KO cells was severely diminished, the infectivity of the limited virus that was released was not reduced [31]. This recapitulated the severe loss in virus particle release found in the IPPK-KO cells of Dick *et al.* [10]. Similarly, production of virus in IPMK-KO cells demonstrated that HIV-1 particles packaged IP6 despite depletion of cellular IP6 and IP5 levels [31]. While these studies show a role for IP6 and IP5, the absolute requirement of these small molecules had not been addressed. Here we demonstrated an absolute requirement for IP6 or IP5 in the production of

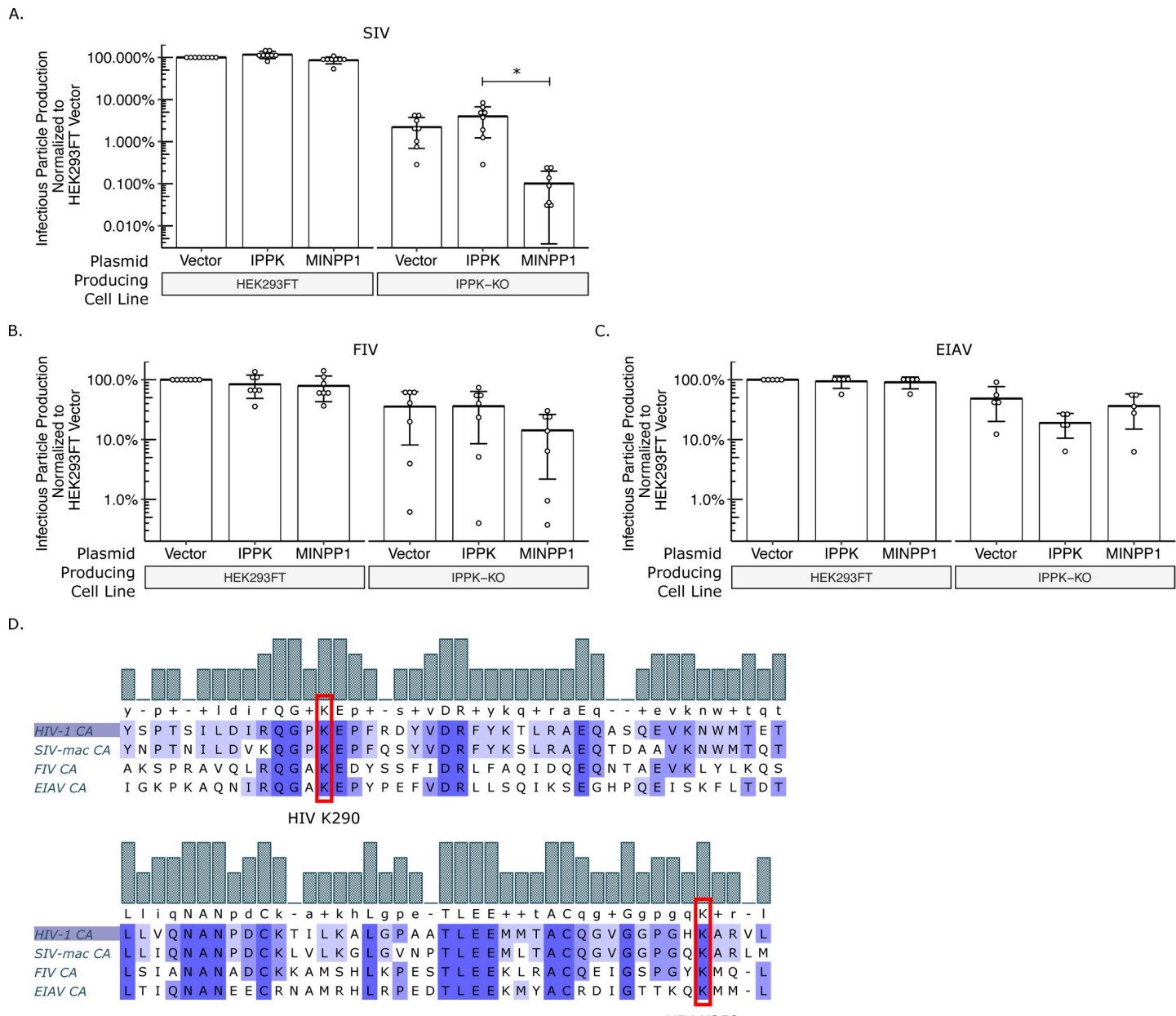

**Fig 8. IP6 and IP5 are required assembly co-factors for primate lentiviruses.** Bar charts of percent infectious particle release of lentiviruses normalized to virus from HEK293FT cells expressing the empty vector. (A) Simian immunodeficiency virus from macaques (SIV, n = 8). (B) Feline immunodeficiency virus (FIV, n = 7). (C) Equine infectious anemia virus (EIAV, n = 5). (D) Multiple sequence alignment of CA proteins of HIV-1, SIV, FIV, and EIAV. Note the homology of K290 and K359 in HIV-1 to lysines in SIV, FIV, and EIAV.

HIV-1 and SIV-mac particles. Furthermore, IP6 and IP5 production in target cells were not required for susceptibility of the cell to infection.

## Knock-out of IPPK and IPMK affects cellular levels of IP6 and IP5 resulting in loss of virus production

We confirmed the ablation of IP6 pools in the IPPK-KO cell line used in Dick *et al.* [10]. There were, however, slightly elevated levels of IP5 in these cells. Since IP5 can substitute for IP6 *in*

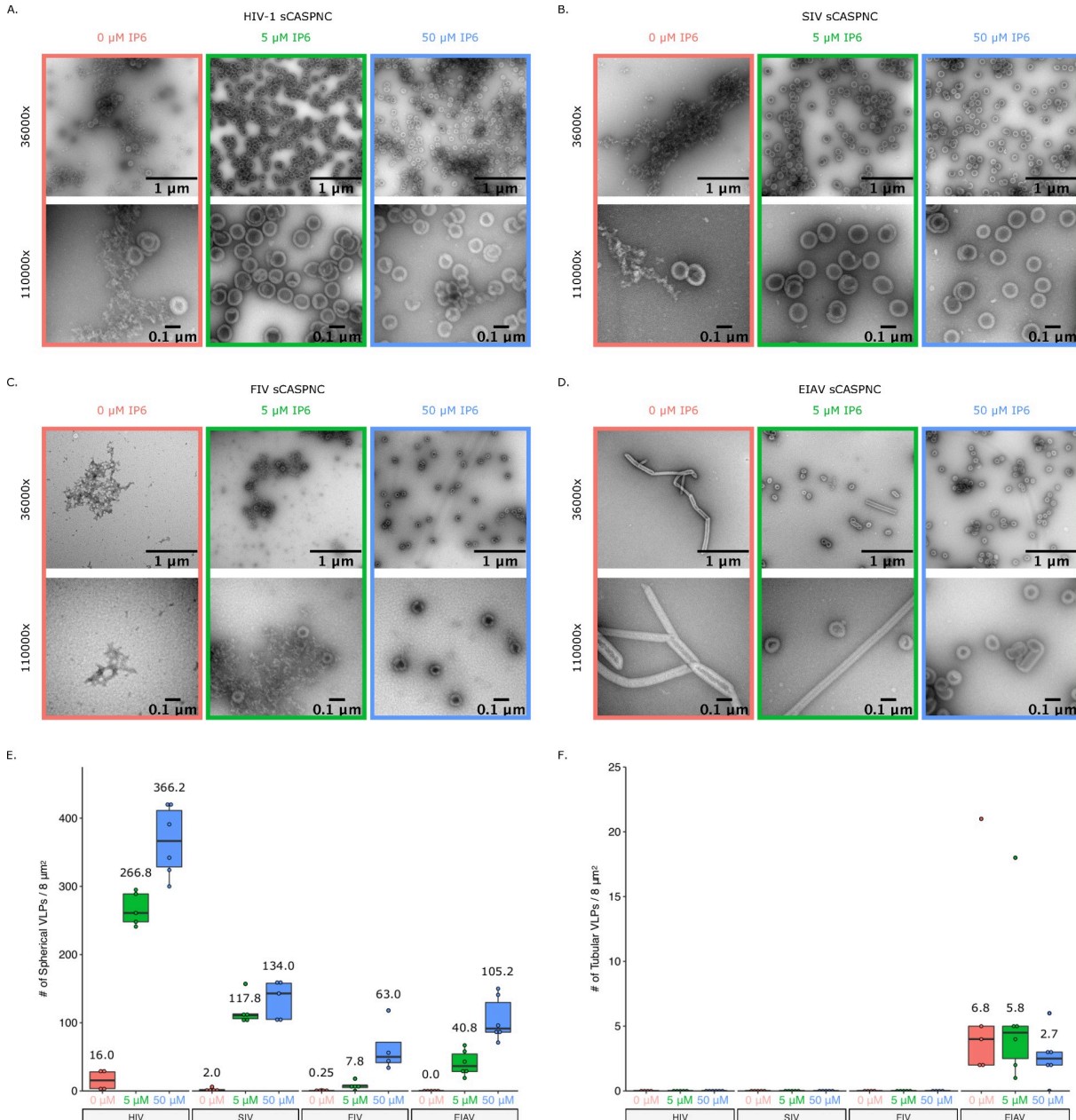

**Fig 9. Addition of IP6 stimulates *in vitro* immature assembly of lentiviruses.** Representative images at 36000x and 110000x of virus like particles (VLPs) from *in vitro* assembly reactions at pH8. Assembly reactions were performed with 0, 5, or 50 μM of IP6. (A) VLPs from HIV-1 sCASPNC (ectopic Serine preceding CASPNC) assemblies. (B) VLPs from SIV sCASPNC assemblies. (C) VLPs from FIV sCACSPNC assemblies. (D) VLPs from EIAV sCASPNC assemblies. (E) Quantification of spherical VLPs from each virus assembly reaction (n = 4–6). (F) Quantification of tubular VLPs from each virus assembly reaction (n = 5–6). Should include a definition of the box plot.

*vitro* and *in vivo* [31], the elevated IP5 could explain the residual virus output from the IPPK-KO cells. Alternatively, IPPK and IP6 have been shown to play roles in many cellular pathways that may have negative effects on virus production. For example, IP6 is involved in the activation of histone deacetylase-1 and mRNA export, with IPPK knock-down causing G1/S phase arrest [41,42]. The IPPK-KO cells proliferate more slowly than HEK293FT cells, in concordance with these data. Thus, ablated IP6 and cell arrest could then negatively impact

HIV-1 transcriptional regulation, by maintenance of acetylated histones or by block of genome export. Conversely, IP6 can promote necroptosis by directly binding mixed lineage kinase domain-like (MLKL) allowing for plasma membrane rupture [22]. IP6 ablation in IPPK-KO cells prevented IP6 direct binding and activation of MLKL, thereby inhibiting necroptosis. Since HIV-1 infection has been shown to mediate necroptosis [43,44], ablation of IP6 may push infected cells toward necroptosis and induce cytopathic effects in neighboring HIV-1 infected cells, thus, reducing overall infectious particle production in the population of cells.

In the IPPK-KO cells, less processed Gag was present in the cell. We believe this is primarily due to the inability of Gag to initiate release from the cell. Protease is activated during the release process and at least some of the processed Gag seen in the cells is budded particles that have remained cell associated or been endocytosed by the cell. The lack of p24 in IPPK-KO cell lysate is consistent with Gag never progressing to the step where protease activation occurs.

In our attempt to identify IP5's role in immature assembly, we knocked-out IPMK using a single guide RNA. The resulting IPMK-KO cell line had residual levels of IP6 and IP5, which correlated with an intermediate loss of virus production. Additionally, residual levels of IP6 and IP5 pointed to an alternative pathway for IP5 synthesis. The role of ITPK1 in inositol phosphate metabolism has not been fully resolved [12,16,21–27]. Since ITPK1 has 5- and 6-kinase activity, this enzyme may compensate for the loss of IPMK (Fig 1A). Attempts to produce IPMK-ITPK1 double knockout cells were not successful likely because of lethality. This led us to take the alternative approach of removing residual levels of IP5 and IP6, instead of preventing their biosynthesis.

## Transient removal of IP6 and IP5 ablates HIV-1 infectious particle production

Inositol phosphates play key roles in cell signaling and modulation of genes regulating these metabolites can have cytopathic effects on the cell as discussed above. We do see slower proliferation of the IPPK-KO cells and cell death with over expression of MINPP1 in agreement with cytopathic effects. While a possible explanation for the assembly and release defects seen in our experiments is the cytotoxicity with mis-regulation of inositol, we believe the phenotype is primarily due to the absence of IP6 and IP5. In support of this conclusion, we see that background production of the house keeping protein GAPDH is relatively similar across treatments (Fig 2F and Fig 5C). In addition, viral protein production in the cell is largely unaffected (Fig 2F and Fig 5C), indicating that the cytopathic effects on the cell do not affect the production of viral proteins. Importantly, other retroviruses (MLV and MPMV) were able to produce and release virus from cells despite IP6/IP5 modulation. Together, these findings provide evidence for IP6/IP5 as the key contributors to the phenotypes we present, not cytopathic effects on the cell.

Transient expression of MINPP1 resulted in substantial loss of IP5 in the IPPK-KO cells. This loss of IP5 correlated with a further decrease in release of infectious virus. This suggests that the residual virus produced from the IPPK-KO cells substituted IP5 for IP6. Mallery *et al*. [31] demonstrated that IP5 can substitute for IP6 and that the released virus is just as infectious per ng of RT as those that utilize IP6; however, they also showed that there was drastic reduction in the amount of virus released. Their data and conclusions additionally showed that Gag is produced from KO cells at similar levels compared to WT, but release of p24 CA is severely impaired from KO cells. The inability of IP6- and IP5-depleted cells to release virus in our transient gene expression system is in agreement with the findings of Mallery *et al* [31] and further provides evidence for an absolute requirement for IP6 or IP5 in immature virus assembly.

While MINPP1 is known to remove only the phosphate at the 3-position on the inositol ring, this position is on the equatorial plane of myo-inositol (Fig 3A) [36–38]. Furthermore, removal of 3-phosphates results in dead-end inositol phosphate species according to the currently known metabolism pathway [12]. These inositol phosphate species are currently not known to be re-phosphorylated to produce relevant IP6 and IP5 for HIV-1 assembly [12]. The negative charge on the equatorial plane is critical for coordinating the lysine ring of the MHR (major homology region) K290 in HIV-1 as demonstrated by the low number of VLPs in assembly reactions with IP4 [10]. It is likely that the residual IP5 detected with MINPP1 addition to IPPK-KO cells corresponds to IP5 species that have equatorial hydroxyls and are not efficiently utilized by lentivirus assembly (Fig 3A).

## Depletion of IP6 and IP5 in target cells does not affect susceptibility to infection

Mature HIV-1 particles use IP6 to stabilize the Fullerene cone capsid structure. IP6 also has been implicated in hexamer pore interactions with dNTPs, required for reverse transcription, and for trafficking to the nuclear envelope [10,29,30,39]. Therefore, it seemed possible that IP6 and IP5 levels could affect these viral interactions during viral entry, and thus cell susceptibility to infection. Here, we demonstrated cells depleted of IP6 and IP5 are just as susceptible to infection as HEK293FT cells. Our data suggest that if IP6 or IP5 are required for viral entry and trafficking, the molecules incorporated during viral assembly are sufficient for this process. We speculate that while cells devoid of IP6 and IP5 cannot produce infectious HIV-1 particles, mutants in Gag might be able to do so. Such mutants might be used to address the dynamics of IP6-capsid interactions at early stages of infection. Furthermore, since inositol phosphates have been implicated in immune responses such as RIG-I signaling [45], there may be signaling responses that can affect the rate of infection. More detailed kinetic studies would be required to investigate this possibility.

## Requirement of IP6 and IP5 is conserved across primate lentiviruses and likely acts as an enhancer for assembly of non-primate lentiviruses

Here, we showed that the requirement for IP6 or IP5 in infectious particles production is a phenotype of primate lentiviruses. *In vitro*, we see stimulatory effects for FIV and EIAV with addition of IP6, which supports the notion that IP6 acts as an enhancer for assembly of non-primate lentiviruses. This may seem at odds with the recent report from Mallery *et al.* [31] where they showed an effect on FIV virus production from IPMK- and IPPK-KO cells. However, that study also found a more pronounced phenotype with HIV-1 than with FIV, and the 3-fold reduction shown with FIV is consistent with the 3-fold decrease that we report here. Further, while we see an enhancement of HIV infectious particle production upon reintroduction of IPPK to the IPPK-KO cell line, we saw no such enhancement with FIV. The Mallery *et al.* study did not explore gene replacement [31]. This suggests that the lower titers with FIV seen in both our data and Mallery et al. (2019) are likely due to cytopathic effects of the IPPK gene KO and not a requirement of IP6/IP5.

We recently showed that EIAV utilizes IP6 as an enhancer for assembly *in vitro* [40]. However, EIAV displayed only a modest 2-fold reduction in infectious particle production from IPPK-KO cells (Fig 8C). As stated earlier, there are cellular cytopathic effects with mis-regulation of inositol phosphates. This likely plays an important role in the reduction we see. Addition of exogenous genes did not further modulate EIAV infectious particle production. While *in vitro*, we see a stimulatory effect in EIAV assembly, there is likely a stricter balance of inositol phosphates required for proper assembly of EIAV *in vivo*.

The robust requirement for IP6 or IP5 is conserved across primate lentiviral species. The lower Gag amino acid sequence homology between lentiviruses and retroviruses of other genera correlates with the lack of a phenotype for beta- and gamma-retroviruses in cells with ablated IP6 and IP5 levels. This suggests that either their structural proteins have relatively stable hexagonal lattice structures and do not require a coordinating molecule, or that another small molecule coordinates structural protein assembly. How viruses evolved to use IP6 is still a topic of great interest and should be further studied.

## Conclusion

In this study, we present data to show that IP6 or IP5 are an absolute requirement for HIV-1 infectious virus particle assembly. Additionally, this robust requirement is likely conserved across primate lentiviruses, but not for other retrovirus genera. While IP6 at high molar concentrations can stimulate *in vitro* assembly for non-primate lentiviruses, the physiological relevance remains to be determined. Furthermore, IP6/IP5 levels in target cells do not affect susceptibility to infection. Understanding at what point IP6 is incorporated into the forming Gag lattice in the cell, for example nucleating assembly of Gag hexamers, may provide new targets for therapeutics.

## Materials and methods

### Plasmid constructs

All lentiviral vectors for CRISPR/Cas-9 delivery were pseudotyped with VSV-g (NIH AIDS Reagent Program) [46]. CRISPR/Cas-9 vectors were derived from the plasmid lentiCRISPRv2 (a gift from Feng Zhang; Addgene plasmid # 52961; http://n2t.net/addgene:52961; RRID: Addgene_52961) [32]. Guide sequences for *IPPK* and *IPMK* were obtained from the Human GeCKOv2 CRISPR knockout pooled libraries (a gift from Feng Zhang; Addgene #1000000048, #1000000049) [32]. Briefly, nucleotide bases as per the lentCRISPRv2 protocol were added to the specific sequences for *IPPK*, *IPMK*, and *ITPK* from the pool, and were cloned into the lentiCRISPRv2 plasmid (Table 1) [32]. The CRISPR/Cas9 vector with guide sequences were

**Table 1. Primers used for cloning.**

| Primer | Sequence |
| --- | --- |
| IPPK guide RNA forward | caccgAACAGCGCTGCGTCGTGCTG |
| IPPK guide RNA reverse | aaacCAGCACGACGCAGCGCTGTTc |
| IPMK guide RNA forward | caccgTCACCTCCCACTGCACCAAA |
| IPMK guide RNA reverse | aaacTTTGGTGCAGTGGGAGGTGAc |
| ITPK1 guide RNA forward | caccgGGCCCTGCTCCTCGATCGGC |
| ITPK1 guide RNA reverse | aaacGCCGATCGAGGAGCAGGGCCc |
| IPPK cDNA forward | cctcgtacgcttaatATGGAAGAGGGGAAGATGGACG |
| IPPK cDNA reverse | gcggaattccggatcTTAGACCTTGTGGAGAACTAATGTGC |
| MINPP1 cDNA forward | cctcgtacgcttaattaaATGCTACGCGCGCCCGGC |
| MINPP1 cDNA reverse | gcggaattccggatccTCATAGTTCATCAGATGTACTG |
| IPPK flanking forward | GAAATGTGTGCCACTGTGTTTA |
| IPPK flanking reverse | ATGATGGACACACCACTTTCT |
| IPMK flanking forward | AGGCTAGAATTAGATAACCAAGAAGAG |
| IPMK flanking reverse | GAGGAAGTCATGCAGAGACAATA |
| ITPK1 flanking forward | CCTGGCCTGTTGACACTATT |
| ITPK1 flanking reverse | GAGCCATTTCTCCAGACTATACC |

delivered via the packaging vector psPAX2 (a gift from Didier Trono; Addgene plasmid # 12260; http://n2t.net/addgene:12260; RRID:Addgene_12260).

All cDNA vectors were packaged with a CMV-MLV-Gag-Pol expression plasmid (kindly provided by Walther Mothes, Yale University) pseudotyped with VSVg (NIH AIDS Reagent Program) [46]. Primers for cDNA amplification were ordered from IDTDNA (Table 1). Primers consisted of 3' sequences matching the coding sequences for *IPPK* and *MINPP1* and had 15 base pair overhangs for InFusion cloning (Clonetech, PT3669-5; Cat. No. 631516) into expression plasmid pQCXIH (Clonetech) using the restriction sites PacI and BamHI. The MINPP1 IRES GFP was made by replacing the hygromycin resistance cassette from the previous clone with IRES-GFP via InFusion cloning.

All retroviruses were pseudotyped with VSV-g (NIH AIDS Reagent Program) [46]. HIV-1$^{\Delta Env}$ consisted of NL4-3-derived proviral vector with a 3' cytomegalovirus (CMV) driven green fluorescent protein (GFP) and defective for Vif, Vpr, Nef, & Env (kindly provided by Vineet Kewal-Ramani, National Cancer Institute-Frederick). HIV-1-CD4 was made by replacing GFP from the previous clone with CD4 via InFusion cloning. SIV$^{\Delta Env}$ consisted of the Gag-Pol expression plasmid pUpSVOΔΨ and the reporter vector plasmid pV1eGFPSVO (kindly provided by Hung Fan, University of California-Irvine). FIV$^{\Delta Env}$ consisted of the Gag-Pol expression plasmid pFP93 and the reporter vector plasmid pGinsin (kindly provided by Eric Poeschla, University of Colorado-Denver) [47]. EIAV$^{\Delta Env}$ consisted of the Gag-Pol expression plasmid pONY3.1 and the reporter vector plasmid pONY8.0-GFP (kindly provided by Nicholas D. Mazarakis, Imperial College-London) [48]. MPMV$^{\Delta Env}$ consisted of an Env-deficient expression plasmid pSARM into which our lab engineered a CMV driven GFP reporter before the 3' LTR (kindly provided by Eric Hunter, Emory University). MLV$^{\Delta Env}$ consisted of the CMV-MLV-Gag-Pol expression plasmid (kindly provided by Walther Mothes, Yale University) and the reporter vector plasmid pQCXIP-GFP (a gift from Michael Grusch (Addgene plasmid # 73014; http://n2t.net/addgene:73014; RRID:Addgene_73014)).

## Cells and knock-out of cellular genes

The HEK293FT cell line was obtained from Invitrogen and maintained in Dulbecco's modified Eagle's medium (DMEM) supplemented with 10% fetal bovine serum (FBS), 2 mM glutamine, 1 mM sodium pyruvate, 10 mM nonessential amino acids, and 1% minimal essential medium vitamins. Knock-out cell lines were obtained via transduction of HEK293FT cells with lenti-CRISPRv2 containing the guide sequences targeting a 5' exon on all known splice variants of the *IPPK* and *IPMK* genes (Table 1). 48 hrs post transduction culture media was replaced with media containing 1 μg/mL puromycin and incubated until complete death of non-transduced control cells (~48–72 hrs). Clonal isolates (> 10) were obtained by sparse plating of surviving cells on a 10 cm dish and allowing colonies to grow. Colonies were then picked and expanded in new plates. KO was verified by amplification of genomic DNA flanking the CRISPR target site (Table 1; 500–600 bp PCR product), direct sequencing of the PCR product, and sequence analysis. Sequences were screened using TIDE (Tracking of Indels by Decomposition; [49]) by comparing Sanger sequence traces of the WT control to the KO. The algorithm not only quantifies major reads, but the underlying background reads and gives p-values for each calculated insertion-deletion. Sequences of clonal isolates were analyzed for mutations that cause frame shifts in the open reading frame that result in premature termination. With the IPPK-KO clone isolates, we saw two distinct sequences diverge after the guide RNA site and is consistent with two alleles. Our lab luckily identified one homozygous clone (a -10 bp deletion) for IPPK-KO. The deletion in this clone causes a premature stop codon immediately upstream (two amino acids) of the guide RNA target site and causes the IPPK gene to be truncated to

~33 amino acids compared to the 492 amino acids of the full length dominate isoform of IPPK. Since the guide RNA targeted a 5' exon of all known splice variants, the other isoforms would be similarly truncated. With the IPMK-KO clonal isolates we saw three distinct sequences diverge from the guide RNA site of our selected IPMK-KO, suggesting three alleles. Such aneuploidy is common in HEK 293 derived cell lines. As can be seen in Fig 1C, there were aberrant sequence traces after the break point at the guide RNA site in the IPMK-KO clone. Deep analysis of these traces showed a -10 bp and -1 bp deletion as well as a +1 bp insertion. As these were not multiples of three, the alleles would have all out-of-frame mutations and would truncate IPMK in a similar manner as the IPPK-KO.

## Virus production and transductions

HIV (NL4-3 derived), SIV (mac), EIAV (pony), MLV (Moloney), VLPs were produced by polyethylenimine (PEI, made in house) transfection of HEK293FTs or KO derivatives at 50% confluence with 900 ng of viral plasmids plus VSV-g in a 9:1 ratio [50]. Media containing virus (viral media) was collected two days post transfection. Viral media was then frozen at -80˚C for a minimum of 1 hr to lyse cells, thawed in a 37˚C water bath, precleared by centrifugation at 3000 x g for 5 min and supernatant collected. Aliquots after titration were stored at -80˚C and subsequently used for assays.

Viral media was titered on HEK293FT cells by serial dilution. Viral media was then added to fresh HEK293FT cells at low MOI to prevent infection saturation. Infected cells were collected and assayed via flow cytometry. Infections were then normalized to percent of infections in WT HEK293FT cells and presented as relative particle production.

## Separation of inositol phosphates

Inositol phosphates were separated and quantified as per Wilson *et al.* [35]. All steps were performed on ice and 100 µL of 100 µM IP6 and 100 µL of 10 µM IP6 standards were treated in parallel as a control. Briefly, inositol phosphates were extracted from counted cells (HEK293FT and IPPK-KO) by suspension in 1 M perchloric acid. Cells were pelleted and supernatants were transferred to tubes containing 4 mg of $TiO_2$ beads (GL Sciences Inc., Titansphere; Cat. No. 5020–75000) in 50 µL 1 M perchloric acid to bind inositol phosphates. Washed inositol phosphates bound to $TiO_2$ beads were eluted with 10% ammonium hydroxide, beads pelleted, and supernatant collected. Supernatant was then concentrated to 10uL and pH neutralized by SpeedVac centrifugation.

A 33% PAGE large gel (16 x 20 cm; TBE pH 8, Acrylamide:Bis 19:1, SDS) was cast and pre-run for 30 min at 500 V. The entirety of concentrated samples (10 µL) was mixed with 50 µL bromophenol blue loading buffer (6X Buffer). The samples were then run over night at 4˚C at 1000 V until the loading dye had traveled through one-third of the gel. The gel was then stained with toluidine blue for 30 min and destained. Gels were then imaged. Molar concentrations were calculated assuming a cell diameter of 15 µM [51]

## Surface labeling of cells

Cells were washed with PBS and treated with 10 mM EDTA. Cells were then collected with PBS, centrifuged at 300 x g for 5 min, and supernatant removed. Cells were blocked with 5% goat serum in PBS for 30 min on ice. Cells were centrifuged at 300 x g for 5 min and supernatant removed. Anti-CD4 antibody conjugated to Alexa-Fluor 555 was applied to cells at 1:100 in 1% goat serum in PBS for 1 hr. Cells were then washed 3 x with PBS, suspended in 400 µL of PBS, and 100 µL of 10% paraformaldehyde (PFA) added to fix cells. After incubation for 20

min on ice, cells were centrifuged at 300 x g for 5 min and supernatant was removed. Cells were resuspended in 200 μL of PBS and analyzed via flow cytometry.

## Flow cytometry

Cells in 6- and 12-well format were washed with PBS and treated with 10 mM TrypLE™ Express Enzyme (Gibco; Cat. No. 12605028). Cells were then collected with PBS and added to 10% PFA to a final concentration of 4%. After 10–20 min incubation at room temperature, the cells were centrifuged at 300 x g for 5 min, supernatant removed, and 300 μL of PBS added. Cells were analyzed for fluorescence using an Accuri C6 flow cytometer.

## Western blot

Supernatant collected after preclearing thawed media containing virus was pelleted via centrifugation through a 20% sucrose cushion (20% sucrose, 100 mM NaCl, 10 mM Tris, 1 mM EDTA, pH 7.5) for 2 h at 30000 x g at 4˚C. Supernatant and sucrose buffer were aspirated off, leaving a small amount of sucrose buffer so as not to aspirate the viral pellet (~10 μL). Ten μL of 2x sample buffer (50 mM Tris, 2% sodium dodecyl sulfate [SDS], 20% glycerol, 5% β-mercaptoethanol) was added to pelleted virus and heated to 95˚C for 5 min before loading.

Cell samples were washed with PBS and trypsinized with 10 mM EDTA. Cells were then collected with PBS, centrifuged at 300 x g for 5 min, and supernatant removed. Twenty μL of RIPA extraction buffer with protease inhibitor was then added to each sample [52]. The samples were then kept on ice and vortexed every 5 min for 20 min, followed by centrifugation at 10000 rpm for 10 min at 4˚C. Supernatant was then transferred to a new tube, 20 μL of 2x sample buffer added, and heated to 95˚C for 5 min before loading.

Samples were separated on a 10% SDS-PAGE gel and transferred onto a 0.22 μm pore size polyvinylidene difluoride (PVDF) membrane. Membranes were blocked for 1 hr at room temperature with 5% nonfat dry milk in PBS-tween. Membranes were then incubated with anti-HIV p24 hybridoma medium was diluted 1:500 (HIV-1 p24 hybridoma [183-H12-5C], obtained from NIH AIDS Reagent Program) from Bruce Chesebro [53] for 1 hr at room temperature. After blots were washed with PBST (3 x for 5 min), horseradish peroxidase (HRP)-conjugated secondary antibody was applied at 1:10,000 to all blots. After 1 hr, blots were again washed 3x with PBST and imaged. Horseradish peroxidase-linked anti-mouse (A5278), was obtained from Sigma. Luminata Classico Western HRP substrate (Millipore) was used for visualization of the membranes with a chemiluminescence image analyzer (UVP BioSpectrum 815 Imaging System).

## Protein purification and in vitro assembly of CANC protein

Protein purification, in vitro assembly, and imaging of all lentiviral CANC proteins was performed as previously described in Dick *et al.* [40] and briefly here. 50 μM protein purified from bacteria was mixed with 10 μM GT25 oligo without or with 5 μM or 10 μM IP6. 30 μL assembly reactions were dialyzed against 2 mL buffer (50 mM Tris-HCl pH 8, 100 mM NaCl, 2 mM TCEP, without or with 5 μM or 10 μM IP6. 4 hrs post dialysis, assembly reactions were adjusted to 200 μL, spotted onto formvar/carbon grids, stained with 2% uranly acetate, and imaged on a FEI Morgagni transmission electron microscope.

## Data analysis

Flow cytometry data was analyzed using FlowJo™ software [54]. Values for fluorescence were exported to Excel spreadsheet. Images were analyzed using Fiji (ImageJ) [55]. Western blot

images were converted to 8-bit, Fiji's gel analysis tools used to calculate density, and values exported to an Excel spreadsheet. VLPs from electron micrographs were quantified manually using Fiji's cell counting tool and values recorded in an Excel spreadsheet. Values from Excel spreadsheets were formatted for statistical analysis via R [56] and exported to CSV format. RStudio was used to analyze data and create figures [57]. UGene was used for plasmid cloning, sequence analysis, and chromatogram image generation [58]. Final figures were prepared using Inkscape [59].

## Acknowledgments

We thank John York for helpful discussions. We thank Volker Vogt for helpful discussions and reviewing the manuscript.

## Author Contributions

**Conceptualization:** Clifton L. Ricana, Marc C. Johnson.

**Data curation:** Clifton L. Ricana, Marc C. Johnson.

**Formal analysis:** Clifton L. Ricana, Marc C. Johnson.

**Funding acquisition:** Marc C. Johnson.

**Investigation:** Clifton L. Ricana, Terri D. Lyddon, Robert A. Dick.

**Methodology:** Clifton L. Ricana, Marc C. Johnson.

**Project administration:** Clifton L. Ricana, Marc C. Johnson.

**Resources:** Robert A. Dick, Marc C. Johnson.

**Supervision:** Marc C. Johnson.

**Validation:** Clifton L. Ricana, Terri D. Lyddon, Robert A. Dick, Marc C. Johnson.

**Visualization:** Clifton L. Ricana, Robert A. Dick.

**Writing – original draft:** Clifton L. Ricana, Marc C. Johnson.

**Writing – review & editing:** Clifton L. Ricana, Robert A. Dick, Marc C. Johnson.

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
