## [Decision Letter · Decision Letter 0]

5 Jun 2020

Dear Dr. Johnson,

Thank you very much for submitting your manuscript "Inositol hexakisphosphate (IP6) and inositol pentakisphosphate (IP5) are required for viral particle release of retroviruses belonging to the primate lentivirus genus" for consideration at PLOS Pathogens. As with all papers reviewed by the journal, your manuscript was reviewed by members of the editorial board and by several independent reviewers. In light of the reviews (below this email), we would like to invite the resubmission of a significantly-revised version that takes into account the reviewers' comments.

We cannot make any decision about publication until we have seen the revised manuscript and your response to the reviewers' comments. Your revised manuscript is also likely to be sent to reviewers for further evaluation.

Sincerely,

Ronald C. Desrosiers

Associate Editor

PLOS Pathogens

Susan Ross

Section Editor

PLOS Pathogens

Kasturi Haldar

Editor-in-Chief

PLOS Pathogens

orcid.org/0000-0001-5065-158X

Michael Malim

Editor-in-Chief

PLOS Pathogens

orcid.org/0000-0002-7699-2064

Reviewer's Responses to Questions

**Part I - Summary**

Reviewer #1: This paper presents data to document the importance of IP6 and IP5 in primate lentivirus particle production. KOs of kinases involved in their synthesis, and overexpression of phosphatases to reduce them, are used to test for effects on virion yield. Previous work has shown that KO of IPPK (no IP6, high IP5) led to large reductions in virus yield. This was confirmed here (Fig 2F). Overexpression of the phosphatase in the KO line, only feasible transiently, reduced IP5 levels to undetectable and further lowered yield to near zero (Fig 5C), indicating that either IP6 or IP5 can support some virus production. Experiments also showed that IP5/6 levels do not matter for the early steps of infection. Non lentivirus, and even nonprimate lentiviruses, do not show the same requirements. Most of the data is clear. There is particularly good analysis and presentation of the IP intermediates and pathways.

Reviewer #2: In this paper, the authors extend previously published work on the role of inositol-pentakisphosphate (IP5) and inositol-hexakisphosphate (IP6) on HIV-1 assembly and particle infectivity. Consistent with a recent publication (ref 31) they report that cells in which inositol-pentakisphosphate 2-kinase (IPPK) has been knocked out express no detectable IP6 but elevated levels of IP5, and display severely reduced virus release but not reduced specific particle infectivity. The authors transiently express multiple inositol polyphosphate phosphatase-1 (MINPP1) in IPPK-KO cells to nearly ablate IP5 and IP6, essentially abolishing virus particle production. The authors also show that HIV-1 infectivity is not significantly reduced when IP5/IP6 are depleted from target cells. Finally, the authors show that primate lentiviruses, but not non-primate lentiviruses or other retroviruses (beta and gamma), are dependent on IP5/IP6 for their assembly.

Characterization of the role of IP6 in HIV-1 assembly and infectivity has been moving at a rapid pace, and the current study provides useful new information. In particular, use of KO cells as targets, and the use of MINPP1, are novel. I have only a couple minor comments for improvement of the MS.

Reviewer #3: Ricana et al. present further evidence as to the importance of IP5/IP6 in HIV particle assembly and release. It has been previously shown that knockout of IPPK which is required to get IP6 results in a reduction in virus production. The question they address here is whether residual IP5 can substitute for the IP6. They are not able to block the production of both IP5 and IP6 in cells as least one of these is required for viability. However, they were able to transiently have cells highly depleted for both IP5 and IP6 by generating a 293 cell line knocked out for IPPK and then further depleting IP5 by transient transfection of the phosphates MINPP1. In these cells, HIV production was abolished. This is the case for SIVmac but not some other lentiviruses and not for MLV suggesting that the requirement is a property of primate lentiviruses.

This is a well written paper with clear figures. The experiments are expertly done and the data are clear and convincing. The study presents a nice complete story.

The main question here is that of significance and novelty. In 2018, in collaboration with the Volker Vogt lab, the authors published an elegant study in Nature showing that IP6 served as a cofactor for HIV assembly and showed cryo-EM showing the molecule sitting at the face of the Gag hexamer structure. In addition, Mallery et al have shown that IP5 can promote assembly of particles in vitro. What’s new here seems to be just that if you totally deplete both IP5 and 6 you get a strong block to virion production. It is novel, but not unexpected. Yet, it is a worthwhile finding as it shows how important it is to have one of these sitting in right place on the hexamer. It’s also significant with respect to drug discovery as it shows that molecules that act on that site of capsid would be valuable. Simply blocking production of these molecules is not going to be possible.

Major points

1. It is hard to get from the paper what the major finding is. It is not fair to conclude that IP5 is required for particle production, as knock-out of the molecule has no effect in normal cells. An accurate statement of their findings might be that in the absence of IP6, IP5 can partially substitute. It is important that the authors make a clear statement as to the point of their study. This should be in the abstract and in the Discussion.

2. Mallery et al. recently showed that IP5 can substitute for IP6. Isn’t that at odds with the results here in which it is shown that IP5 only poorly substitutes for IP6? The authors need to clarify whether this is a discrepancy or can be explained.

3. One wonders whether the transient depletion of IP5/6 from the cells is having other effects that lead to the failure to assemble. These molecules play critical signaling roles in cells. Are the authors sure that the assembly deficiency is due to the absence of the molecules from the virion and not from another effect on the cell?

4. Fig. 2F. IPPK knock-out prevents virus release. In these cells there should be a buildup of Gag Pr55. In fact, the cells show less Gag precursor. That would suggest that it’s not a block to assembly, but a block to Gag synthesis.

5. The earlier Mallery et al. Cell Reports paper shows that FIV requires IP6. That seems to be at odds with the results here. The authors need to comment on this apparent discrepancy.

**Part II – Major Issues: Key Experiments Required for Acceptance**

Reviewer #1: Major points:

The tests for the effects on virion production are minimal here (really it’s all Fig 5C). We probably need error bars on this key experiment.

Some of the presentation is a little confusing in its relationship to the earlier work. As the authors point out (line 106), “Knockouts of the IPPK and IPMK genes have both been reported to reduce the production of infectious HIV-1 particles in vivo.” Thus, work very similar to what we see here is in the literature (Cell Rep. 2019;29(12):3983 and Nature. 2018;560: 509). Still, there are some aspects of the present study that are new: We do get a deeper understanding of the relative importance of IP5 vs. IP6. And issues such as requirements for virus entry are addressed. Nonprimate viruses are examined. The paper acknowledges the earlier work, but doesn’t make it totally clear what is old news and what is truly new information – we need a sharper transition.

I think we need to be given more data on the nature of the KO mutations being studied here. 293 cells are notoriously aneuploid, and often have 3 or 4 or more copies of various chromosomes. Thus the KO clones generated need to be characterized to know that all alleles were genuinely mutated, to know what the alleles are, and to be sure that no wild-type alleles or partially active alleles are still lurking. This all becomes more crucial when there is the potential for partial activity to explain the findings. On line 429 of methods – how many clones were tested to find good KOs, and how many clones were tested for virus (I gather only one)? How many alleles were examined in each cell clone? What were the alleles – any in-frame mutations? “Direct sequencing” of PCR products is not adequate, because it only gives the majority allele. There is a broad assumption here that CRISPR is perfect every time, and this is definitely not true.

Reviewer #2: N/A

Reviewer #3: none

**Part III – Minor Issues: Editorial and Data Presentation Modifications**

Reviewer #1: Minor points:

1. While the MINPP1 overexpression after transient transfection was apparently successful in eliminating detectable IP5 and IP6, giving a window to examine effects on virus, there is always the concern that the cells are potentially very sick – as indicated by the inability to recover stable transfectants. Given this situation, how confident can we be that the inhibition of virus production is not a problem with the general metabolic state of the cells, caused by the low IP5/6? One pretty good answer is the near normal virion release of MLV and MPMV. This point could be made more forcefully.

2. In the title, maybe viral particle “release” is the wrong word – it probably isn’t release per se that requires these compounds. Better would be viral particle “production” or something else. This is true elsewhere (l. 155 etc.).

3. In the abstract, line 31, one of the middle sentences would be clearer if it were changed to something like “…but transient expression of the enzyme multiple inositol polyphosphate

phosphatase-1 (MINPP1) in IPPK-KOs….”

4. line 154ff: It’s probably too hard to measure low levels of CA, but it would be nice if the level of physical particles could be quantitatively compared with infectious virus, to confirm that the low levels of released virus are of normal infectivity. It’s certainly roughly true.

5. line 281: “neither virus” is unclear. The authors mean EIAV and FIV, but since in the last sentence MLV and MPMV are also mentioned, this needs to be reworded.

6. Some comments on the processing of Gag might be warranted. In the KO, Gag is apparently no longer processed. Would a protease minus Gag behave any differently in terms of virion yield in any of the experiments? Some speculation could be included.

Reviewer #2: 1. As mentioned above, several key findings, e.g., the ability of IP5 to substitute for IP6 in virus assembly, and the lack of an infectivity defect in virus produced from IPPK KO cells (which unfortunately was not properly tested in the authors’ Nature paper because they measured particle infectivity, not particle assembly/release), were made in ref 31 but this seems to be a bit downplayed here.

2. The EIAV results would appear to contrast with what these authors recently published (Dick et al., PLoS Pathog 2020). This should be clarified for the reader.

Reviewer #3: 1. Fig. 2. B and C. Remove shading from histogram bars.

2. Fig. 2 E. Error bars should be made more visible.

3. Fig. 4B. typo on “transduce” should be “transduced”.

4. Fig. 5B. the log scale on the Y axis showing the tick marks looks weird. Also in other figures. Remove this.

PLOS authors have the option to publish the peer review history of their article (what does this mean?). If published, this will include your full peer review and any attached files.

Reviewer #1: No

Reviewer #2: No

Reviewer #3: No
---

## [Editor Report · Decision Letter 1]

26 Jun 2020

Dear Dr. Johnson,

We are pleased to inform you that your manuscript 'Primate lentiviruses require Inositol hexakisphosphate (IP6) or inositol pentakisphosphate (IP5) for the production of viral particles' has been provisionally accepted for publication in PLOS Pathogens.

Best regards,

Ronald C. Desrosiers

Associate Editor

PLOS Pathogens

Susan Ross

Section Editor

PLOS Pathogens

Kasturi Haldar

Editor-in-Chief

PLOS Pathogens

orcid.org/0000-0001-5065-158X

Michael Malim

Editor-in-Chief

PLOS Pathogens

orcid.org/0000-0002-7699-2064
---

## [Editor Report · Acceptance letter]

29 Jul 2020

Dear Dr. Johnson,

We are delighted to inform you that your manuscript, "Primate lentiviruses require Inositol hexakisphosphate (IP6) or inositol pentakisphosphate (IP5) for the production of viral particles," has been formally accepted for publication in PLOS Pathogens.

Best regards,

Kasturi Haldar

Editor-in-Chief

PLOS Pathogens

orcid.org/0000-0001-5065-158X

Michael Malim

Editor-in-Chief

PLOS Pathogens

orcid.org/0000-0002-7699-2064